# Iterative energy-based projection on a normal data manifold for anomaly localization

**David Dehaene,**[*] **Oriel Frigo,**[*] **Sébastien Combrexelle, Pierre Eline**
AnotherBrain, Paris, France
`{david, oriel, sebastien, pierre}@anotherbrain.ai`

## Abstract

Autoencoder reconstructions are widely used for the task of unsupervised anomaly localization. Indeed, an autoencoder trained on normal data is expected to only be able to reconstruct normal features of the data, allowing the segmentation of anomalous pixels in an image via a simple comparison between the image and its autoencoder reconstruction. In practice however, local defects added to a normal image can deteriorate the whole reconstruction, making this segmentation challenging. To tackle the issue, we propose in this paper a new approach for projecting anomalous data on a autoencoder-learned normal data manifold, by using gradient descent on an energy derived from the autoencoder's loss function. This energy can be augmented with regularization terms that model priors on what constitutes the user-defined optimal projection. By iteratively updating the *input* of the autoencoder, we bypass the loss of high-frequency information caused by the autoencoder bottleneck. This allows to produce images of higher quality than classic reconstructions. Our method achieves state-of-the-art results on various anomaly localization datasets. It also shows promising results at an inpainting task on the CelebA dataset.

## 1 Introduction

Automating visual inspection on production lines with artificial intelligence has gained popularity and interest in recent years. Indeed, the analysis of images to segment potential manufacturing defects seems well suited to computer vision algorithms. However these solutions remain data hungry and require knowledge transfer from human to machine via image annotations. Furthermore, the classification in a limited number of user-predefined categories such as non-defective, greasy, scratched and so on, will not generalize well if a previously unseen defect appears. This is even more critical on production lines where a defective product is a rare occurrence. For visual inspection, a better-suited task is unsupervised anomaly detection, in which the segmentation of the defect must be done only via prior knowledge of non-defective samples, constraining the issue to a two-class segmentation problem.

From a statistical point of view, an anomaly may be seen as a distribution outlier, or *an observation that deviates so much from other observations as to arouse suspicion that it was generated by a different mechanism* (Hawkins, 1980). In this setting, generative models such as Variational AutoEncoders (VAE, Kingma & Welling (2014)), are especially interesting because they are capable to infer possible sampling mechanisms for a given dataset. The original autoencoder (AE) jointly learns an encoder model, that compresses input samples into a low dimensional space, and a decoder, that decompresses the low dimensional samples into the original input space, by minimizing the distance between the input of the encoder and the output of the decoder. The more recent variant, VAE, replaces the deterministic encoder and decoder by stochastic functions, enabling the modeling of the distribution of the dataset samples as well as the generation of new, unseen samples. In both models, the output decompressed sample given an input is often called the *reconstruction*, and is used as some sort of projection of the input on the support of the normal data distribution, which we will call the *normal manifold*. In most unsupervised anomaly detection methods based on VAE, models are trained on flawless data and defect detection and localization is then performed using a

---

[*]Equal contributions.

distance metric between the input sample and its reconstruction (Bergmann et al., 2018; 2019; An & Cho, 2015; Baur et al., 2018; Matsubara et al., 2018).

One fundamental issue in this approach is that the models learn on the normal manifold, hence there is no guarantee of the generalization of their behavior outside this manifold. This is problematic since it is precisely outside the dataset distribution that such methods intend to use the VAE for anomaly localization. Even in the case of a model that always generates credible samples from the dataset distribution, there is no way to ensure that the reconstruction will be connected to the input sample in any useful way. An example illustrating this limitation is given in figure 1, where a VAE trained on regular grid images provides a globally poor reconstruction despite a local perturbation, making the anomaly localization challenging.

In this paper, instead of using the VAE reconstruction, we propose to find a better projection of an input sample on the normal manifold, by optimizing an energy function defined by an autoencoder architecture. Starting at the input sample, we iterate gradient descent steps on the input to converge to an optimum, simultaneously located on the data manifold and closest to the starting input. This method allows us to add prior knowledge about the expected anomalies via regularization terms, which is not possible with the raw VAE reconstruction. We show that such an optimum is better than previously proposed autoencoder reconstructions to localize anomalies on a variety of unsupervised anomaly localization datasets (Bergmann et al., 2019) and present its inpainting capabilities on the CelebA dataset (Liu et al., 2015). We also propose a variant of the standard gradient descent that uses the pixel-wise reconstruction error to speed up the convergence of the energy.

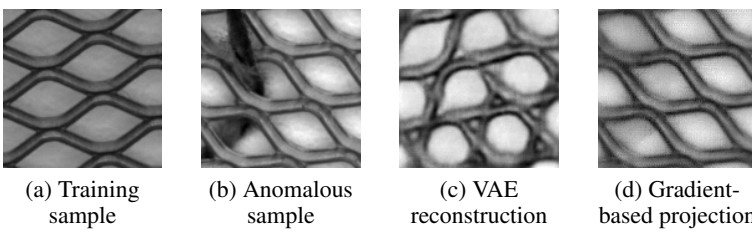

| (a) Training sample | (b) Anomalous sample | (c) VAE reconstruction | (d) Gradient-based projection |

Figure 1: Even though an anomaly is a local perturbation in the image (b), the whole VAE-reconstructed image can be disturbed (c). Our gradient descent-based method gives better quality reconstructions (d).

## 2 BACKGROUND

### 2.1 GENERATIVE MODELS

In unsupervised anomaly detection, the only data available during training are samples $\mathbf{x}$ from a non-anomalous dataset $\mathbb{X} \subset \mathbb{R}^d$. In a generative setting, we suppose the existence of a probability function of density $q$, having its support on all $\mathbb{R}^d$, from which the dataset was sampled. The generative objective is then to model an estimate of density $q$, from which we can obtain new samples close to the dataset. Popular generative architectures are Generative Adversarial Networks (GAN, Goodfellow et al. (2014)), that concurrently train a generator $G$ to generate samples from random, low-dimensional noise $\mathbf{z} \sim p$, $\mathbf{z} \in \mathbb{R}^l$, $l \ll d$, and a discriminator $D$ to classify generated samples and dataset samples. This model converges to the equilibrium of the expectation over both real and generated datasets of the binary cross entropy loss of the classifier $min_G \, max_D \, [\, \mathbb{E}_{\mathbf{x} \sim q} \, [\log(D(\mathbf{x}))] \, + \, \mathbb{E}_{\mathbf{z} \sim p} \, [\log(1 \, - \, D(G(\mathbf{z})))] \,]$.

Disadvantages of GANs are that they are notoriously difficult to train (Goodfellow, 2017), and they suffer from mode collapse, meaning that they have the tendency to only generate a subset of the original dataset. This can be problematic for anomaly detection, in which we do not want some subset of the normal data to be considered as anomalous (Bergmann et al., 2019). Recent works such as Thanh-Tung et al. (2019) offer simple and attractive explanations for GAN behavior and propose substantial upgrades, however Ravuri & Vinyals (2019) still support the point that GANs have more trouble than other generative models to cover the whole distribution support.

Another generative model is the VAE (Kingma & Welling (2014)), where, similar to a GAN generator, a decoder model tries to approximate the dataset distribution with a simple latent variables prior $p(\mathbf{z})$, with $\mathbf{z} \in \mathbb{R}^l$, and conditional distributions output by the decoder $p(\mathbf{x}|\mathbf{z})$. This leads to the estimate $p(\mathbf{x}) = \int p(\mathbf{x}|\mathbf{z})p(\mathbf{z})dz$, that we would like to optimize using maximum likelihood estimation on the dataset. To render the learning tractable with a stochastic gradient descent (SGD) estimator with reasonable variance, we use importance sampling, introducing density functions $q(\mathbf{z}|\mathbf{x})$ output by an encoder network, and Jensen's inequality to get the variational lower bound :

$$
\begin{aligned}
\log p(\mathbf{x}) = \log \ \mathbb{E}_{\mathbf{z}\sim q(\mathbf{z}|\mathbf{x})} \frac{p(\mathbf{x}|\mathbf{z})p(\mathbf{z})}{q(\mathbf{z}|\mathbf{x})} \\
\geq \mathbb{E}_{\mathbf{z}\sim q(\mathbf{z}|\mathbf{x})} \log p(\mathbf{x}|\mathbf{z}) - D_{\mathrm{KL}}(q(\mathbf{z}|\mathbf{x})\|p(\mathbf{z})) = -\mathcal{L}(\mathbf{x})
\end{aligned}
\tag{1}
$$

We will use $\mathcal{L}(\mathbf{x})$ as our loss function for training. We define the VAE reconstruction, per analogy with an autoencoder reconstruction, as the deterministic sample $f_{VAE}(\mathbf{x})$ that we obtain by encoding $\mathbf{x}$, decoding the mean of the encoded distribution $q(\mathbf{z}|\mathbf{x})$, and taking again the mean of the decoded distribution $p(\mathbf{x}|\mathbf{z})$.

VAEs are known to produce blurry reconstructions and generations, but Dai & Wipf (2019) show that a huge enhancement in image quality can be gained by learning the variance of the decoded distribution $p(\mathbf{x}|\mathbf{z})$. This comes at the cost of the distribution of latent variables produced by the encoder $q(\mathbf{z})$ being farther away from the prior $p(\mathbf{z})$, so that samples generated by sampling $\mathbf{z} \sim p(\mathbf{z}), \mathbf{x} \sim p(\mathbf{x}|\mathbf{z})$ have poorer quality. The authors show that using a second VAE learned on samples from $q(\mathbf{z})$, and sampling from it with ancestral sampling $\mathbf{u} \sim p(\mathbf{u}), \mathbf{z} \sim p(\mathbf{z}|\mathbf{u}), \mathbf{x} \sim p(\mathbf{x}|\mathbf{z})$, allows to recover samples of GAN-like quality. The original autoencoder can be roughly considered as a VAE whose encoded and decoded distributions have infinitely small variances.

## 2.2 ANOMALY DETECTION AND LOCALIZATION

We will consider that an anomaly is a sample with low probability under our estimation of the dataset distribution. The VAE loss, being a lower bound on the density, is a good proxy to classify samples between the anomalous and non-anomalous categories. To this effect, a threshold $T$ can be defined on the loss function, delimiting anomalous samples with $\mathcal{L}(\mathbf{x}) \geq T$ and normal samples $\mathcal{L}(\mathbf{x}) < T$. However, according to Matsubara et al. (2018), the regularization term $\mathcal{L}_{KL}(\mathbf{x}) = D_{\mathrm{KL}}(q(\mathbf{z}|\mathbf{x})\|p(\mathbf{z}))$ has a negative influence in the computation of anomaly scores. They propose instead an unregularized score $\mathcal{L}_r(\mathbf{x}) = -\mathbb{E}_{\mathbf{z}\sim q(\mathbf{z}|\mathbf{x})} \log p(\mathbf{x}|\mathbf{z})$ which is equivalent to the reconstruction term of a standard autoencoder and claim a better anomaly detection.

Going from anomaly detection to anomaly localization, this reconstruction term becomes crucial to most of existing solutions. Indeed, the inability of the model to reconstruct a given *part* of an image is used as a way to segment the anomaly, using a pixel-wise threshold on the reconstruction error. Actually, this segmentation is very often given by a pixel-wise (An & Cho, 2015; Baur et al., 2018; Matsubara et al., 2018) or patch-wise comparison of the input image, and some generated image, as in Bergmann et al. (2018; 2019), where the structural dissimilarity (DSSIM, Wang et al. (2004)) between the input and its VAE reconstruction is used.

Autoencoder-based methods thus provide a straightforward way of generating an image conditioned on the input image. In the GAN original framework, though, images are generated from random noise $\mathbf{z} \sim p(\mathbf{z})$ and are not conditioned by an input. Schlegl et al. (2017) propose with AnoGAN to get the closest generated image to the input using gradient descent on $\mathbf{z}$ for an energy defined by:

$$
E_{AnoGAN} = ||\mathbf{x} - G(\mathbf{z})||_1 + \lambda \cdot ||f_D(\mathbf{x}) - f_D(G(\mathbf{z}))||_1
\tag{2}
$$

The first term ensures that the generation $G(\mathbf{z})$ is close to the input $\mathbf{x}$. The second term is based on a distance between features of the input and the generated images, where $f_D(\mathbf{x})$ is the output of an intermediate layer of the discriminator. This term ensures that the generated image stays in the vicinity of the original dataset distribution.

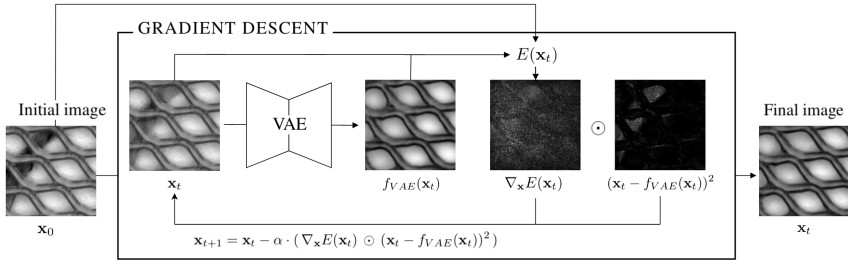

Figure 2: Illustration of our method. We perform gradient descent on $E(\mathbf{x}_t)$ to iteratively correct $\mathbf{x}_t$.

# 3 PROPOSED METHOD

## 3.1 ADVERSARIAL PROJECTIONS

According to Zimmerer et al. (2018), the loss gradient with respect to $\mathbf{x}$ gives the direction towards normal data samples, and its magnitude could indicate how abnormal a sample is. In their work on anomaly identification, they use the loss gradient as an anomaly score.

Here we propose to use the gradient of the loss to iteratively improve the observed $\mathbf{x}$. We propose to link this method to the methodology of computing adversarial samples in Szegedy et al. (2014).

After training a VAE on non-anomalous data, we can define a threshold $T$ on the reconstruction loss $\mathcal{L}_r$ as in (Matsubara et al., 2018), such that a small proportion of the most improbable samples are identified as anomalies. We obtain a binary classifier defined by

$$A(\mathbf{x}) = \begin{cases} 1 \text{ if } \mathcal{L}_r(\mathbf{x}) \geq T \\ 0 \text{ otherwise} \end{cases} \tag{3}$$

Our method consists in computing adversarial samples of this classifier (Szegedy et al., 2014), that is to say, starting from a sample $\mathbf{x}_0$ with $A(\mathbf{x}_0) = 1$, iterate gradient descent steps over the input $\mathbf{x}$, constructing samples $\mathbf{x}_1, \dots \mathbf{x}_N$, to minimize the energy $E(\mathbf{x})$, defined as

$$E(\mathbf{x}_t) = \mathcal{L}_r(\mathbf{x}_t) + \lambda \cdot ||\mathbf{x}_t - \mathbf{x}_0||_1 \tag{4}$$

An iteration is done by calculating $\mathbf{x}_{t+1}$ as

$$\mathbf{x}_{t+1} = \mathbf{x}_t - \alpha \cdot \nabla_{\mathbf{x}} E(\mathbf{x}_t), \tag{5}$$

where $\alpha$ is a learning rate parameter, and $\lambda$ is a parameter trading off the inclusion of $\mathbf{x}_t$ in the normal manifold, given by $\mathcal{L}_r(\mathbf{x}_t)$, and the proximity between $\mathbf{x}_t$ and the input $\mathbf{x}_0$, assured by the regularization term $||\mathbf{x}_t - \mathbf{x}_0||_1$.

## 3.2 REGULARIZATION TERM

We model the anomalous images that we encounter as normal images in which a region or several regions of pixels are altered but the rest of the pixels are left untouched. To recover the best segmentation of the anomalous pixels from an anomalous image $\mathbf{x}_a$, we want to recover the closest image from the normal manifold $\mathbf{x}_g$. The term *closest* has to be understood in the sense that the smallest number of pixels are modified between $\mathbf{x}_a$ and $\mathbf{x}_g$. In our model, we therefore would like to use the $L^0$ distance as a regularization distance of the energy. Since the $L^0$ distance is not differentiable, we use the $L^1$ distance as an approximation.

## 3.3 OPTIMIZATION IN INPUT SPACE

While in our method the optimization is done in the input space, in the previously mentioned AnoGAN, the search for the optimal reconstruction is done by iterating over $\mathbf{z}$ samples with the

energy defined in equation 2. Following the aforementioned analogy between a GAN generator $G$ and a VAE decoder $Dec$, a similar approach in the context of a VAE would be to use the energy

$$||\mathbf{x} - Dec(\mathbf{z})||_1 - \lambda \cdot \log p(\mathbf{z}) \qquad (6)$$

where the $-\log p(\mathbf{z})$ term has the same role as AnoGAN's $||f_D(\mathbf{x}) - f_D(G(\mathbf{z}))||_1$ term, to ensure that $Dec(\mathbf{z})$ stays within the learned manifold. We chose not to iterate over $\mathbf{z}$ in the latent space for two reasons. First, because as noted in Dai & Wipf (2019) and Hoffman & Johnson (2016), the prior $p(\mathbf{z})$ is not always a good proxy for the real image of the distribution in the latent space $q(\mathbf{z})$. Second, because the VAE tends to ignore some details of the original image in its reconstruction, considering that these details are part of the independent pixel noise allowed by the modeling of $p(\mathbf{x}|\mathbf{z})$ as a diagonal Gaussian, which causes its infamous blurriness. An optimization in latent space would have to recreate the high frequency structure of the image, whereas iterating over the input image space, and starting the descent on the input image $\mathbf{x}_0$, allows us to keep that structure and thus to obtain projections of higher quality.

## 3.4 OPTIMIZING GRADIENT DESCENT

We observed that using the Adam optimizer (Kingma & Ba, 2015) is beneficial for the quality of the optimization. Moreover, to speed up the convergence and further preserve the aforementioned high frequency structure of the input, we propose to compute our iterative samples using the pixel-wise reconstruction error of the VAE. To explain the intuition behind this improvement, we will consider the inpainting task. In this setting, as in anomaly localization, a local perturbation is added on top of a normal image. However, in the classic inpainting task, the localization of the perturbation is known beforehand, and we can use the localization mask $\mathbf{\Omega}$ to only change the value of the anomalous pixels in the gradient descent:

$$\mathbf{x}_{t+1} = \mathbf{x}_t - \alpha \cdot (\nabla_{\mathbf{x}} E(\mathbf{x}_t) \odot \mathbf{\Omega}) \qquad (7)$$

where $\odot$ is the Hadamard product.

For anomaly localization and blind inpainting, where this information is not available, we compute the pixel-wise reconstruction error which gives a rough estimate of the mask. The term $\nabla_{\mathbf{x}} E(\mathbf{x}_t)$ is therefore replaced with $\nabla_{\mathbf{x}} E(\mathbf{x}_t) \odot (\mathbf{x}_t - f_{VAE}(\mathbf{x}_t))^2$ in equation 5:

$$\mathbf{x}_{t+1} = \mathbf{x}_t - \alpha \cdot (\nabla_{\mathbf{x}} E(\mathbf{x}_t) \odot (\mathbf{x}_t - f_{VAE}(\mathbf{x}_t))^2) \qquad (8)$$

where $f_{VAE}(\mathbf{x})$ is the standard reconstruction of the VAE. Optimizing the energy this way, a pixel where the reconstruction error is high will update faster, whereas a pixel with good reconstruction will not change easily. This prevents the image to update its pixels where the reconstruction is already good, even with a high learning rate. As can be seen in appendix B, this method converges to the same performance as the method of equation 5, but with fewer iterations. An illustration of our method can be found in figure 2.

## 3.5 STOP CRITERION

A standard stop criterion based on the convergence of the energy can efficiently be used. Using the adversarial setting introduced in section 3.1, we also propose to stop the gradient descent when a certain predefined threshold on the VAE loss is reached. For example, such a threshold can be chosen to be a quantile of the empirical loss distribution computed on the training set.

## 4 EXPERIMENTS

In this section, we evaluate the proposed method for two different applications: anomaly segmentation and image inpainting. Both applications are interesting use cases of our method, where we search to reconstruct partially corrupted images, correcting the anomalies while preserving the uncorrupted image regions.

## 4.1 UNSUPERVISED ANOMALY SEGMENTATION

In order to evaluate the proposed method for the task of anomaly segmentation, we perform experiments with the recently proposed MVTec dataset (Bergmann et al., 2019). This collection of datasets

Table 1: Results for anomaly segmentation on MVTec datasets, expressed in AUROC (Area Under the Receiver Operating Characteristics). Four different baselines are trained on normal samples and are augmented by our proposed gradient based reconstruction (grad) for comparison: A deterministic autoencoder trained with $L^2$ loss ($L^2$AE) as in (Bergmann et al., 2019); A deterministic autoencoder trained with DSSIM loss (DSAE) as in (Bergmann et al., 2019); A variational autoencoder (VAE); And a variational autoencoder with a learned decoder variance ($\gamma$-VAE) as in (Dai & Wipf, 2019). For each result a green or red background denotes respectively an improvement or a decrease in performance compared to the baseline. It can be seen that the proposed gradient-based reconstruction achieves the best segmentation for most datasets, with a *mean improvement rate of 9.52% over all baselines*.

| | Category | $L^2$AE | $L^2$AE grad | DSAE | DSAE grad | VAE | VAE grad | $\gamma$-VAE | $\gamma$-VAE grad |
|---|---|---|---|---|---|---|---|---|---|
| Textures | carpet | 0.539 | 0.734 | 0.545 | **0.774** | 0.580 | 0.735 | 0.648 | 0.727 |
| | grid | 0.960 | **0.981** | 0.960 | 0.980 | 0.888 | 0.961 | 0.950 | 0.979 |
| | leather | 0.751 | 0.921 | 0.710 | 0.602 | 0.834 | **0.925** | 0.818 | 0.897 |
| | tile | 0.476 | 0.575 | 0.496 | 0.626 | 0.465 | **0.654** | 0.491 | 0.581 |
| | wood | 0.630 | 0.805 | 0.641 | 0.738 | 0.695 | **0.838** | 0.665 | 0.809 |
| Objects | bottle | 0.909 | 0.916 | 0.933 | **0.951** | 0.902 | 0.922 | 0.913 | 0.931 |
| | cable | 0.732 | 0.864 | 0.790 | 0.859 | 0.828 | **0.910** | 0.777 | 0.880 |
| | capsule | 0.786 | **0.952** | 0.769 | 0.884 | 0.862 | 0.917 | 0.814 | 0.917 |
| | hazelnut | 0.976 | 0.984 | 0.966 | 0.966 | 0.977 | 0.976 | 0.977 | **0.988** |
| | metalnut | 0.880 | 0.899 | 0.881 | **0.920** | 0.881 | 0.907 | 0.883 | 0.914 |
| | pill | 0.885 | 0.912 | 0.895 | 0.927 | 0.888 | 0.930 | 0.897 | **0.935** |
| | screw | 0.979 | 0.980 | **0.983** | 0.925 | 0.958 | 0.945 | 0.976 | 0.972 |
| | toothbrush | 0.971 | 0.983 | 0.973 | 0.984 | 0.971 | **0.985** | 0.971 | 0.983 |
| | transistor | 0.906 | 0.921 | 0.904 | **0.934** | 0.894 | 0.919 | 0.896 | 0.931 |
| | zipper | 0.680 | **0.889** | 0.828 | 0.887 | 0.814 | 0.869 | 0.706 | 0.871 |

consists of 15 different categories of objects and textures in the context of industrial inspection, each category containing a number of normal and anomalous samples.

We train our model on normal training samples and test it on both normal and anomalous test samples to evaluate the anomaly segmentation performance.

We perform experiments with three different baseline autoencoders: A "vanilla" variational autoencoder with decoder covariance matrix fixed to identity (Kingma & Welling, 2014), a variational autoencoder with learned decoder variance (Dai & Wipf, 2019), a "vanilla" deterministic autoencoder trained with $L^2$ as reconstruction loss ($L^2$AE) and a deterministic autoencoder trained with DSSIM reconstruction loss (DSAE), as proposed by Bergmann et al. (2018). For the sake of a fair comparison, all the autoencoder models are parameterized by convolutional neural networks with the same architecture, latent space dimensionality (set to 100), learning rate (set to 0.0001) and number of epochs (set to 300). The architecture details (layers, paddings, strides) are the same as described in Bergmann et al. (2018) and Bergmann et al. (2019). Similarly to the authors in Bergmann et al. (2019), for the textures datasets, we first subsample the original dataset images to $512 \times 512$ and then crop random patches of size $128 \times 128$ which are used to train and test the different models. For the object datasets, we directly subsample the original dataset images to $128 \times 128$ unlike in Bergmann et al. (2019) who work on $256 \times 256$ images, then we perform rotation and translation data augmentations. For all datasets we train on 10000 images.

Anomaly segmentation is then computed by reconstructing the anomalous image and comparing it with the original. We perform the comparison between reconstructed and original with the DSSIM metric as it has been observed in Bergmann et al. (2018) that it provides better anomaly localization than $L^2$ or $L^1$ distances. For the gradient descent, we set the step size $\alpha := 0.5$, $L^1$ regularization weight $\lambda := 0.05$ and the stop criterion is achieved when a sample reconstruction loss is inferior to the minimum reconstruction loss over the training set.

In table 1 we show the AUROC (Area Under the Receiver Operating Characteristics) for different autoencoder methods, with different thresholds applied to the DSSIM anomaly map computed be-

tween original and reconstructed images. Note that an AUROC of 1 expresses the best possible segmentation in terms of normal and anomalous pixels. For each autoencoder variant we compare the baseline reconstruction with the proposed gradient-based reconstruction (grad.). As in Bergmann et al. (2019) we observe that an overall best model is hard to identify, however we show that our method increases the AUC values for almost all autoencoder variants. Aggregating the results over all datasets and baselines, we report a mean improvement rate of 9.52%, with a median of 4.33%, a 25th percentile of 1.86%, and a 75th percentile of 15.86%. The histogram of the improvement rate for all datasets and baselines is provided in appendix F, as well as a short analysis.

In figure 3 we compare our anomaly segmentation with a baseline $L^2$ autoencoder Bergmann et al. (2019) ($L^2$AE) for a number of image categories. For all results in figure 3, we set the same threshold to 0.2 to the anomaly detection map given by the DSSIM metric. The visual results in figure 3 highlights an overall improvement of anomaly localization by our proposed iterative reconstruction ($L^2$AE-grad). See appendix C for additional visual results of anomaly segmentation on remaining categories of MVTec dataset, and on remaining baseline models.

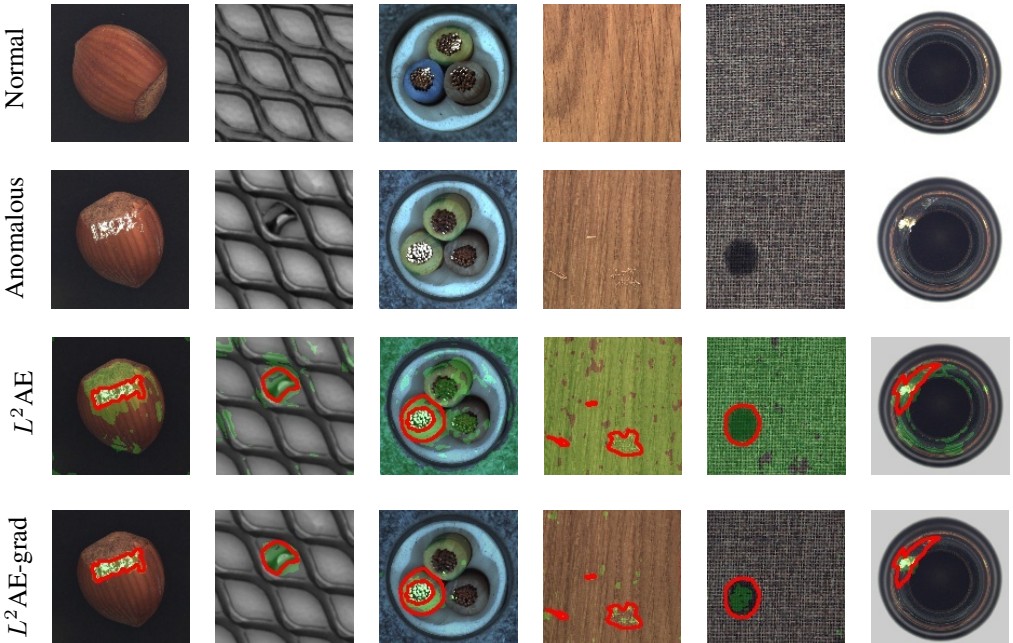

Figure 3: First row: Normal samples of hazelnut, grid, cable, wood, carpet and bottle categories in MVTec dataset; Second row: anomalous samples from the aforementioned dataset categories; Third row: Anomaly segmentation with baseline $L^2$ autoencoder (Bergmann et al., 2019); Fourth row: our proposed anomaly segmentation with $L^2$ autoencoder augmented with gradient-based iterative reconstruction. Ground truth is represented by red contour, and each estimated segmentation by a green overlay. It can be seen that anomaly segmentation is refined by our proposed method, with a tendency of detecting less false positives.

## 4.2 INPAINTING

Image inpainting is a well known image reconstruction problem which consists of reconstructing a corrupted or missing part of an image, where the region to be reconstructed is usually given by a known mask. Many different approaches for inpainting have been proposed in the literature, such as anisotropic diffusion (Bertalmio et al., 2000), patch matching (Criminisi et al., 2004), context autoencoders (Pathak et al., 2016) and conditional variational autoencoders (Ivanov et al., 2019).

If we consider that the region to be reconstructed is not known beforehand, the problem is sometimes called *blind inpainting* (Altinel et al., 2018), and the corrupted part can be seen as an anomaly to be corrected.

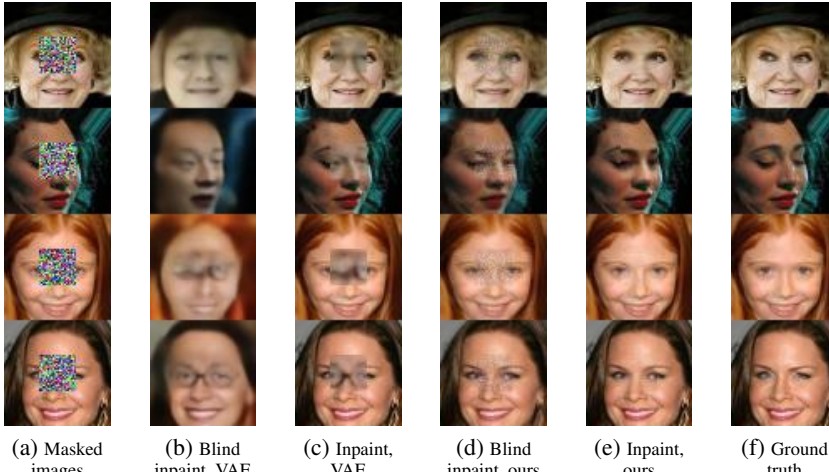

| (a) Masked images | (b) Blind inpaint, VAE | (c) Inpaint, VAE | (d) Blind inpaint, ours | (e) Inpaint, ours | (f) Ground truth |

Figure 4: Inpainting experiment performed on CelebA dataset, where the test face images are masked with uniform noise. The baseline VAE reconstruction is disturbed by the noise mask, providing a poor inpainting. The proposed gradient-based VAE provides a more convincing inpainting by an iterative process.

We performed experiments with image inpainting on the CelebA dataset (Liu et al., 2015), which consists of celebrity faces. In figure 4 we compare the inpainting results obtained with a baseline VAE with learned variance ($\gamma$-VAE) and Resnet architecture, as described by Dai & Wipf (2019), with the same VAE model, augmented by our proposed gradient-based iterative reconstruction. Note that for the regular inpainting task, gradients are multiplied by the inpainting mask at each iteration (equation 7), while for the blind inpainting task, the mask is unknown. See appendix D for a comparison with a recent method based on variational autoencoders, proposed by Ivanov et al. (2019).

## 5 RELATED WORK

Baur et al. (2018) have used autoencoder reconstructions to localize anomalies in MRI scans, and have compared several variants using diverse per-pixel distances as well as perceptual metrics derived from a GAN-like architecture. Bergmann et al. (2018) use the structural similarity metric (Wang et al., 2004) to compare the original image and its reconstruction to achieve better anomaly localization, and also presents the SSIM autoencoder, which is trained directly with this metric.

Zimmerer et al. (2018) use the derivative of the VAE loss function with respect to the input, called the *score*. The amplitude of the score is supposed to indicate how abnormal a pixel is. While we agree that the gradient of the loss is an indication of an anomaly, we think that we have to integrate this gradient over the path from the input to the normal manifold to obtain meaningful information. We compare our results to score-based results for anomaly localization in appendix A.

The work that is the most related to ours is AnoGAN (Schlegl et al., 2017). We have mentioned above the differences between the two approaches, which, apart from the change in underlying architectures, boil down to the ability in our method to update directly the input image instead of searching for the optimal latent code. This enables the method to converge faster and above all to keep higher-frequency structures of the input, which would have been deteriorated if it were passed through the AE bottleneck. Bergmann et al. (2019) compare standard AE reconstructions techniques to AnoGAN, and observes that AnoGAN's performances on anomaly localizations tasks are poorer than AE's due to the mode collapse tendency of GAN architectures. Interestingly, updates on AnoGAN such as fast AnoGAN (Schlegl et al., 2019) or AnoVAEGAN (Baur et al., 2018) replaced the gradient descent search of the optimal **z** with a learned encoder model, yielding an approach very similar to the standard VAE reconstruction-based approaches, but with a reconstruction loss learned by a discriminator, which is still prone to mode collapse (Thanh-Tung et al., 2019).

## 6 CONCLUSION

In this paper, we proposed a novel method for unsupervised anomaly localization, using gradient descent of an energy defined by an autoencoder reconstruction loss. Starting from a sample under test, we iteratively update this sample to reduce its autoencoder reconstruction error. This method offers a way to incorporate human priors into what is the optimal projection of an out-of-distribution sample into the normal data manifold. In particular, we use the pixel-wise reconstruction error to modulate the gradient descent, which gives impressive anomaly localization results in only a few iterations. Using gradient descent in the input data space, starting from the input sample, enables us to overcome the autoencoder tendency to provide blurry reconstructions and keep normal high frequency structures. This significantly reduces the number of pixels that could be wrongly classified as defects when the autoencoder fails to reconstruct high frequencies. We showed that this method, which can easily be added to any previously trained autoencoder architecture, gives state-of-the-art results on a variety of unsupervised anomaly localization datasets, as well as qualitative reconstructions on an inpainting task. Future work can focus on replacing the $L^1$-based regularization term with a Bayesian prior modeling common types of anomalies, and on further improving the speed of the gradient descent.

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

# A COMPARISON WITH ZIMMERER ET AL. (2019)

Table 2: Complementary results for anomaly segmentation on MVTec datasets, expressed in AU-ROC for different pixel-wise scores derived from a baseline VAE: from left to right, squared error reconstruction $||\mathbf{x} - f_{VAE}(\mathbf{x})||^2$ (denoted $\mathcal{L}_r(\mathbf{x})$), gradient of the loss $|\nabla_{\mathbf{x}}\mathcal{L}(\mathbf{x})|$, combination of both, gradient of the KL divergence $|D_{\mathrm{KL}}(q(\mathbf{z}|\mathbf{x})\|p(\mathbf{z}))$ (denoted $|\nabla_{\mathbf{x}}\mathcal{L}_{KL}(\mathbf{x})|$) as well as combination of KL derivative and error reconstruction as suggested in Zimmerer et al. (2018; 2019).

|  | Category | $\mathcal{L}_r(\mathbf{x})$ | $\|\nabla_{\mathbf{x}}\mathcal{L}(\mathbf{x})\|$ | $\|\nabla_{\mathbf{x}}\mathcal{L}(\mathbf{x})\| \odot \mathcal{L}_r(\mathbf{x})$ | $\|\nabla_{\mathbf{x}}\mathcal{L}_{KL}(\mathbf{x})\|$ | $\|\nabla_{\mathbf{x}}\mathcal{L}_{KL}(\mathbf{x})\| \odot \mathcal{L}_r(\mathbf{x})$ | VAE-grad |
|---|---|---|---|---|---|---|---|
| **Textures** | carpet | 0.537 | 0.580 | 0.566 | 0.553 | 0.555 | **0.735** |
|  | grid | 0.823 | 0.635 | 0.812 | 0.507 | 0.790 | **0.961** |
|  | leather | 0.783 | 0.650 | 0.792 | 0.627 | 0.791 | **0.925** |
|  | tile | 0.547 | 0.606 | 0.581 | 0.623 | 0.588 | **0.654** |
|  | wood | 0.686 | 0.691 | 0.726 | 0.643 | 0.713 | **0.838** |
| **Objects** | bottle | 0.831 | 0.762 | 0.832 | 0.629 | 0.830 | **0.922** |
|  | cable | 0.831 | 0.796 | 0.846 | 0.674 | 0.841 | **0.910** |
|  | capsule | 0.765 | 0.754 | 0.772 | 0.642 | 0.795 | **0.917**. |
|  | hazelnut | 0.907 | 0.831 | 0.908 | 0.468 | 0.885 | **0.976** |
|  | metalnut | 0.833 | 0.831 | 0.870 | 0.710 | 0.834 | **0.907** |
|  | pill | 0.869 | 0.833 | 0.872 | 0.480 | 0.826 | **0.930** |
|  | screw | 0.851 | 0.726 | 0.842 | 0.412 | 0.795 | **0.945** |
|  | toothbrush | 0.942 | 0.798 | 0.943 | 0.619 | 0.939 | **0.985** |
|  | transistor | 0.788 | 0.843 | 0.834 | 0.801 | 0.836 | **0.919** |
|  | zipper | 0.725 | 0.674 | 0.729 | 0.562 | 0.727 | **0.869** |

Zimmerer et al. (2019) proposed to perform anomaly localization using different scores derived from the gradient of the VAE loss. In particular, it has been shown that the product of the VAE reconstruction error with the gradient of the KL divergence was very informative for medical images. In table 2 we compare the pixel-wise anomaly detection AUROC of these different scores with our method. For all experiments, we use the same "vanilla" VAE as described in section 4.1.

It can be seen that other VAE-based methods using a single evaluation of the gradient are constantly outperformed by our method.

## B  CONVERGENCE SPEED

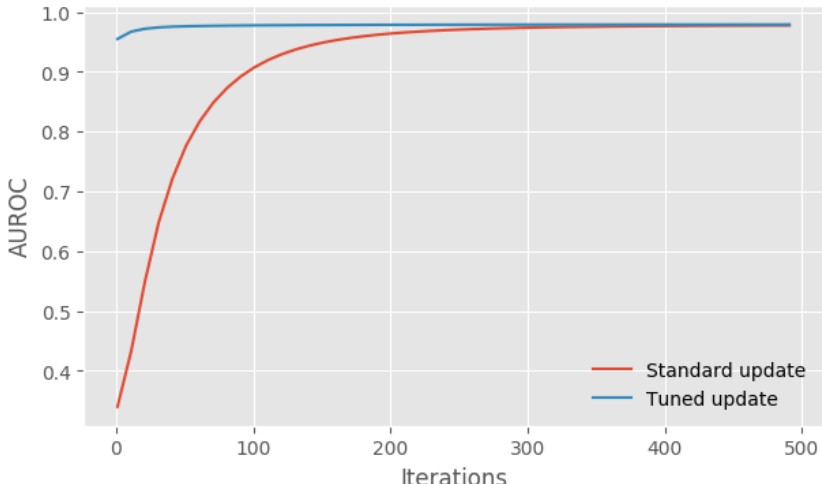

Figure 5: Evolution of pixel-wise anomaly detection AUROC performance.

In figure 5 we compare the number of iterations needed to reach convergence with our two proposals for gradient descent: Standard update as in equation 5 and Tuned update using a gradient mask computed with the VAE reconstruction error, as in equation 8. The model is a VAE with learned decoder variance (Dai & Wipf, 2019), trained on the Grid dataset (Bergmann et al., 2019). We compute the mean pixel-wise anomaly detection AUROC after each iteration on the test set.

We can see that the tuned method converges to the same performance as the standard method, with far fewer iterations.

# C  ADDITIONAL ANOMALY SEGMENTATION RESULTS

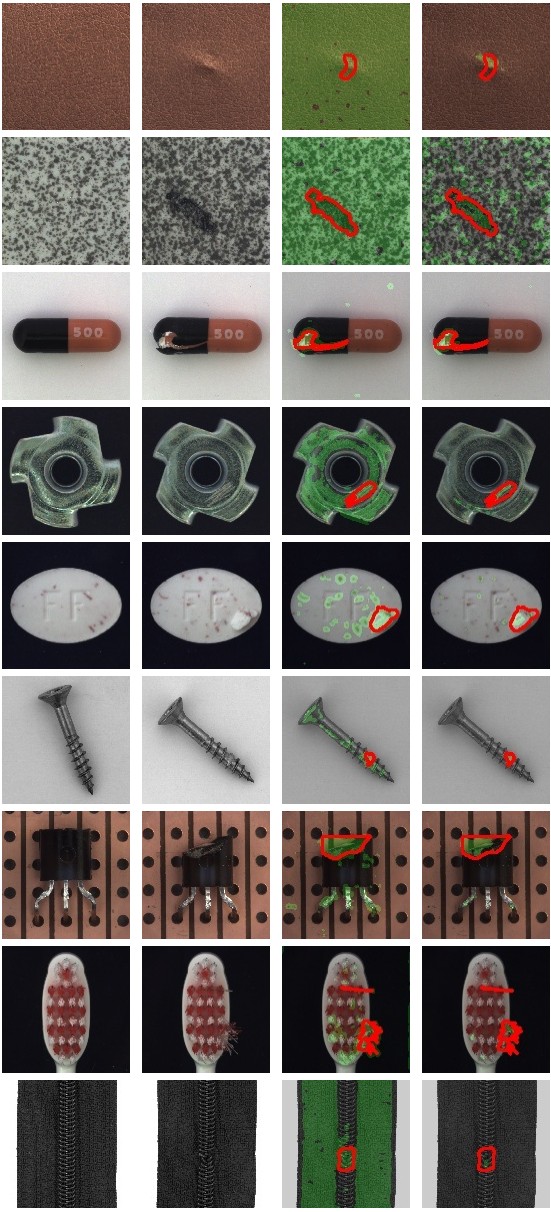

Figure 6: From left to right: Normal; Anomalous; Anomaly segmentation with baseline $L^2$ autoencoder (Bergmann et al., 2019); Our proposed anomaly segmentation with $L^2$ autoencoder augmented with gradient-based iterative reconstruction.

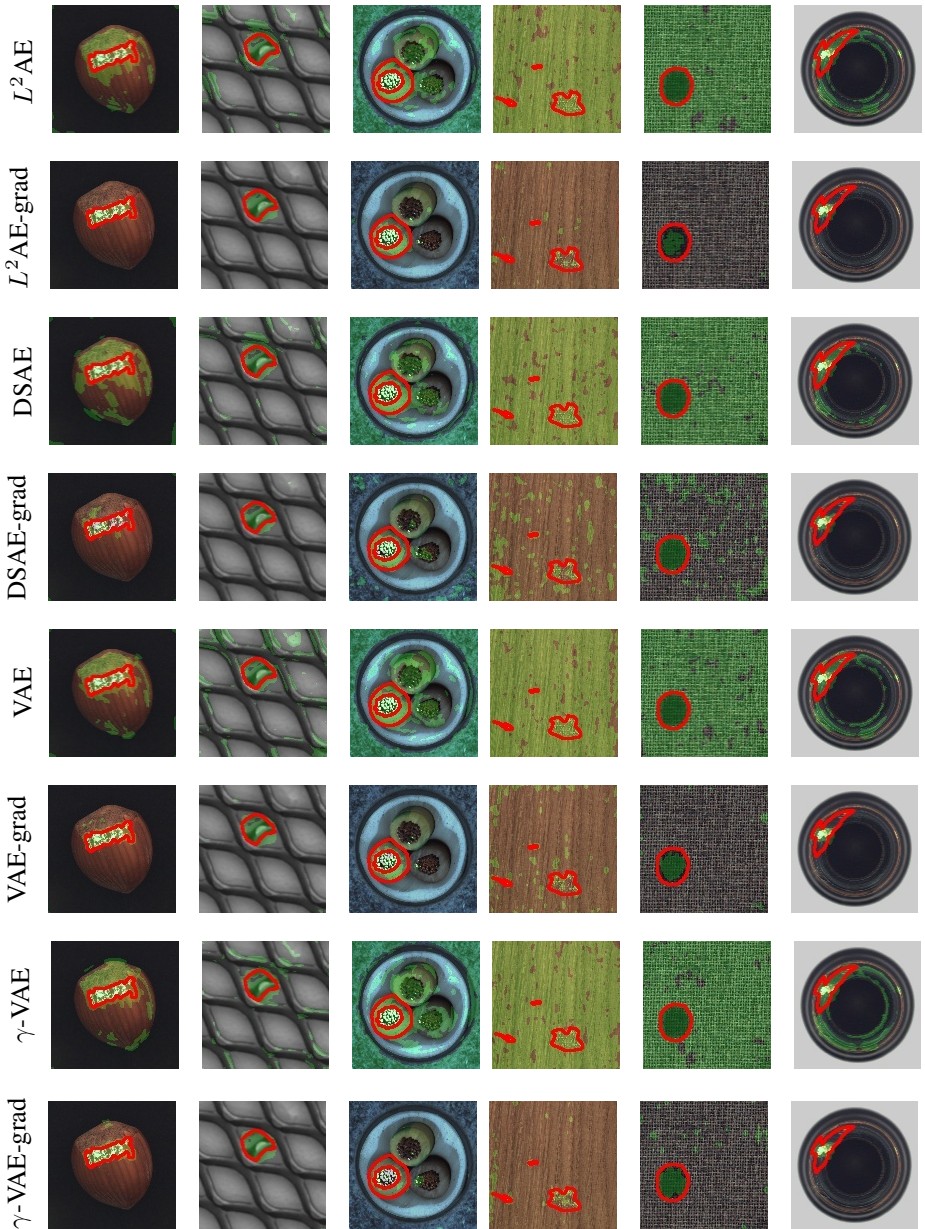

Figure 7: Illustration of anomaly localization comparison over four baselines ($L^2$AE, DSAE, VAE, $\gamma$-VAE). Ground truth is represented by red contour, and each estimated segmentation by a green overlay. It can be seen that anomaly segmentation is overall improved when different baselines are augmented by our proposed gradient descent.

# D INPAINTING COMPARISON

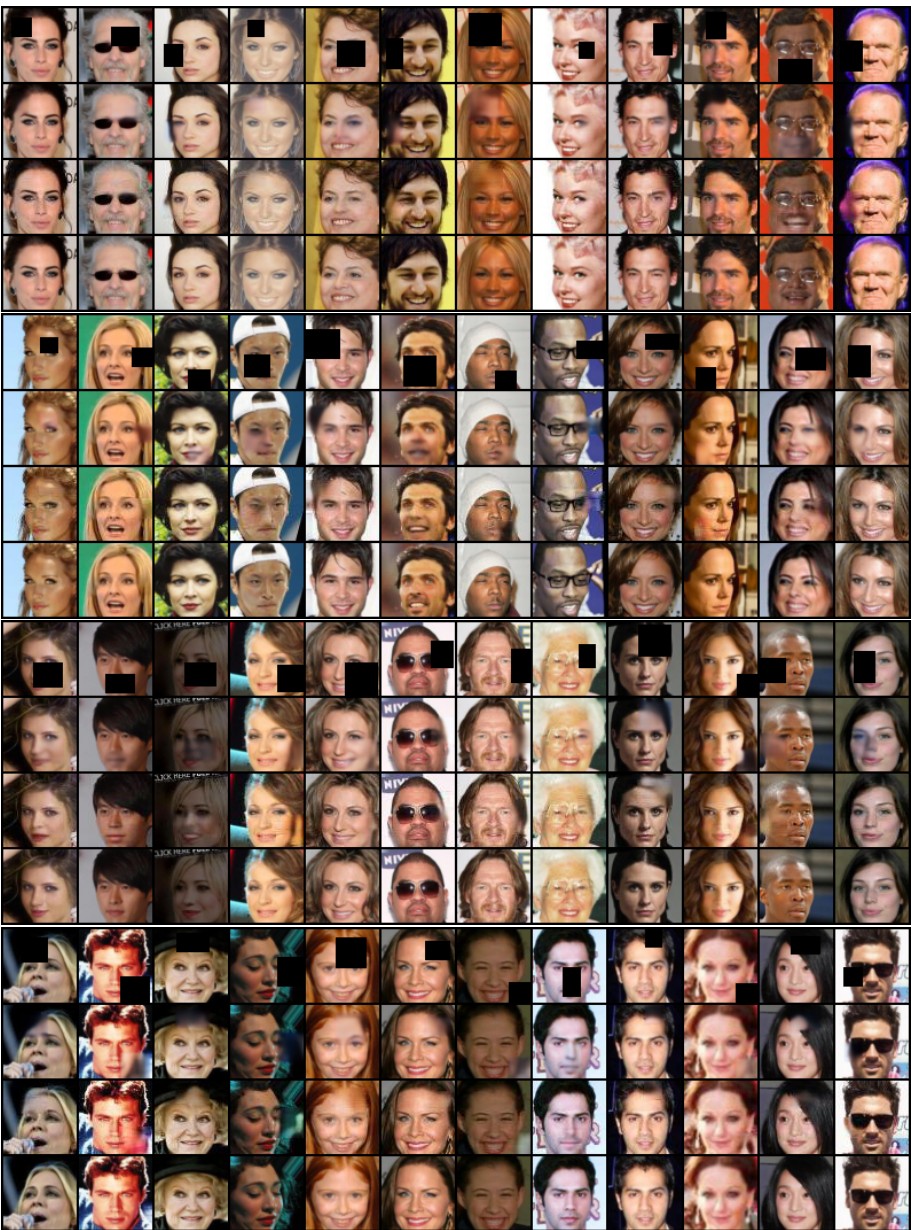

Figure 8: Inpainting comparison. Each batch is made of four rows, from top to bottom: Masked input image; VAE with arbitrary conditioning (VAEAC, Ivanov et al. (2019)); Ours; Ground truth. The quality of the reconstructions is comparable, even though our VAE is trained without any assumptions over the mask's properties.

# E ILLUSTRATION OF THE OPTIMIZATION PROCESS

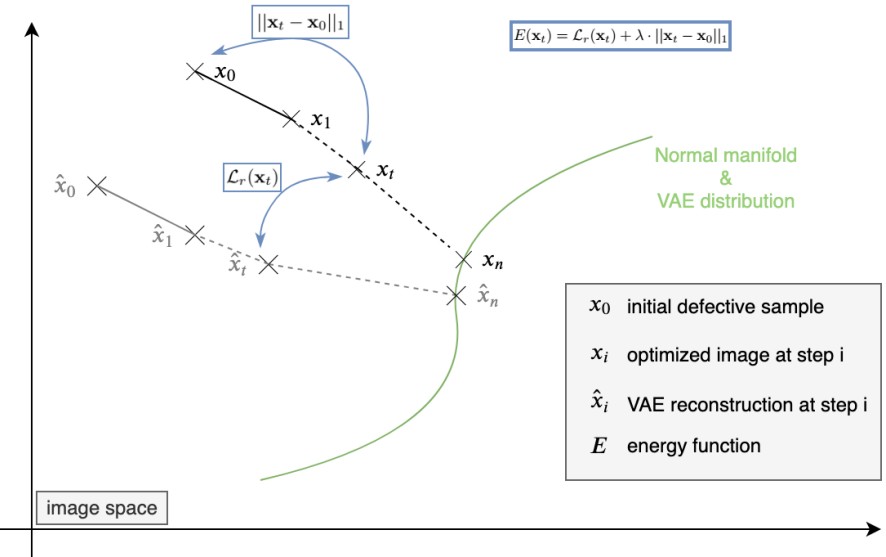

Figure 9: Principle of the energy optimization to project anomalous sample on the normal manifold

Figure 9 illustrates our method principle. We start with a defective input $x_0$ whose reconstruction $\hat{x}_0$ does not necessarily lie on the normal data manifold. As the optimization process carries on, the optimized sample $x_0$ and its reconstruction look more similar and get closer to the manifold. The regularization term of the energy function makes sure that the optimized sample stays close to the original sample.

## F    DISTRIBUTION OF THE IMPROVEMENT RATE ON MVTEC AD

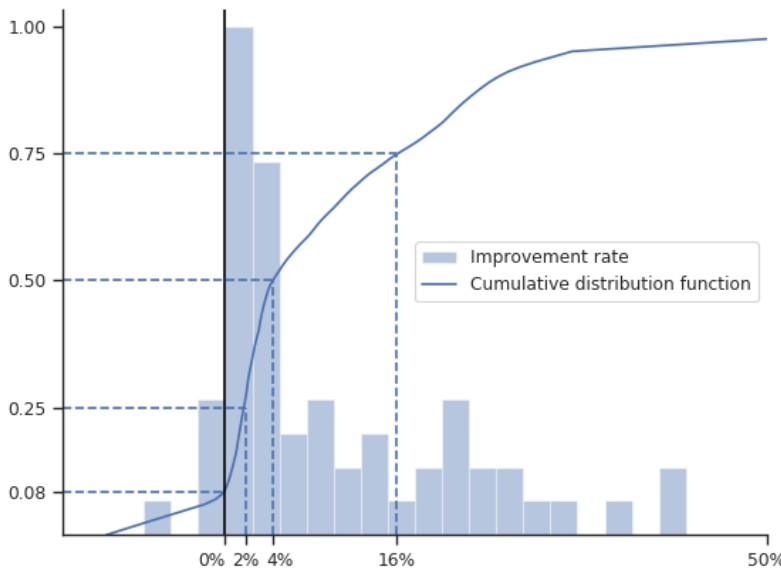

Figure 10: Distribution of the improvement rate over all presented baselines and all datasets in MVTec AD.

Figure 10 shows the distribution of the AUC improvement rate over all presented baselines and all datasets in MVTec AD using our gradient-based projection method.

$$\text{improvement rate} = \frac{AUC_{grad} - AUC_{base}}{AUC_{base}}$$

- 8.3% of data points are under the 0 value delimiting an increase or decrease in AUC due to our method, and 91.7% data points are over this value. Our method increases the AUC in a vast majority of cases.
- The median is at 4.33%, the 25th percentile at 1.86%, and the 75th percentile at 15.86%.

