# OpenReview forum: "Iterative energy-based projection on a normal data manifold for anomaly localization"
_ICLR.cc/2020/Conference — Accept (Poster)_

### Official Review · AnonReviewer3 · 2019-10-27
**Official Blind Review #3**

**Rating:** 3

**Review:**

Summary: The paper proposes to use autoencoder for anomaly localization. The approach learns to project anomalous data on an autoencoder-learned manifold by using gradient descent on energy derived from the autoencoder's loss function. The proposed method is evaluated using the anomaly-localization dataset (Bergmann et al. CVPR 2019) and qualitatively for the task of image inpainting task on CelebA dataset.


Pros:

+ surprisingly simple approach that led to significantly better results.

+ applications to image inpainting, and demonstrates better visual results than using simple VAE.

Concern :

- While I agree that authors have shown relative performance compared to various approaches,  I am not able to map the results of Table-1 to that of Table-3 (second column ROC values) in Bergmann et al. CVPR'19. The setup in two works seem similar. Can the authors please comment to help me understand this difference?

- The proposed approach leads to better performance over the baseline models; it is not clear what is a suitable baseline model for the problem of anomaly localization is?

- The results for image inpainting looks promising. The authors may want to add comparison with existing image inpainting approaches for the reader to better appreciate the proposed approach.

**Experience Assessment:**

I do not know much about this area.

**Review Assessment: Checking Correctness Of Derivations And Theory:**

I carefully checked the derivations and theory.

**Review Assessment: Checking Correctness Of Experiments:**

I carefully checked the experiments.

**Review Assessment: Thoroughness In Paper Reading:**

I read the paper at least twice and used my best judgement in assessing the paper.

---

> ### Author Response · Authors · 2019-11-08
> **Reply to Reviewer3**
>
> Dear reviewer, thank you for your time and comments. First, we would like to draw your attention to our  general comment, answering questions about overall baselines for anomaly localization and statistics on the benefits of our method. This comment also explains changes in the last revision of the paper.
>
> We will address your concerns in order :
>
> 1) We understand your concern about the differences in AUROC values compared to Bergmann et al. CVPR'19.
> Please note that as the code for Bergmann et al. CVPR'19 has not been released, we have implemented ourselves the L2 and DSSIM autoencoder baselines used in the paper. However, our experimental settings have some differences which may explain why the empirical AUC values are not exactly the same of Bergmann et al. CVPR'19. These differences are motivated by a desire to have a single setup of architecture and hyperparameters for all datasets. In details :
>
> - As we explained in the Section 4.1 of our paper, we always work with images of size 128×128 for textures and objects, while Bergmann et al. work with images of size 256×256 for objects datasets and 128×128 for textures datasets.
> - The exact parameters for the random translation and rotation data augmentations for objects datasets are not provided in the paper of Bergmann et al. CVPR'19, thus it is very likely that we do not perform exactly the same data augmentation and thus the training data for the models may be different.
> - We always compute the ROC directly patch-by-patch from the autoencoder anomaly maps, while Bergmann et al. CVPR'19 reconstruct the texture images from the fusion of overlapping patches and perform averaging of the resulting anomaly maps.
>
> We computed plots to compare Bergmann et al. results with our baselines.
> L2-AE :
> https://i.imgur.com/RENg0yG.png
> SSIM-AE:
> https://i.imgur.com/pocIdPw.png
> These show that while there are indeed differences in the two implementations, the trend remains comparable. Furthermore, as detailed in our general comment, we feel that the main contribution of our paper is a method to improve the results of any AE-based model rather than providing a new baseline model for anomaly detection.
>
> 2) Considering the MVTec anomaly dataset, we could argue that there is not a single baseline model which has the best performance for every object and texture category. This is what we observed in our experiments, and it goes in line with the experiments in the benchmark performed by Bergmann et al. 2019. Nevertheless, the autoencoder trained with L2 or DSSIM has the best performance in average in Bergmann et al. 2019, so we included these two baselines. Most importantly, we showed that most of the time AE baselines, deterministic or probabilistic, have a performance improvement when augmented by our method.
> We have also provided in our general comment statistics of the improvement rate due to our method over all presented baselines and datasets.
>
> 3) With respect to the inpainting evaluation, we have provided in Appendix D a qualitative comparison with the recent work of Ivanov et al, ICLR 2019. The quality of the reconstructions is comparable, even though our VAE is trained without any assumptions over the mask's properties. This comparison was not included in the main text for lack of space.
>
> We hope that we have answered your concerns, thank you again.

---

### Official Review · AnonReviewer1 · 2019-10-28
**Official Blind Review #1**

**Rating:** 6

**Review:**

This paper discusses an important problem of solving the visual inspection problem limited supervision.  It proposes to use VAE to model the anomaly detection. The major concern is how the quality of f_{VAE} is estimated. From the paper it seems f_{VAE} is not updated. Will it be sufficient to rely a fixed f_{VAE} and blindly trust its quality?

Detailed Comments:
- Table 1: It is not clear how "the mean improvement rate of 9.52% over all baselines" was calculated.
- Figure 3: Will VAE-grad or DASE-grad perform better? Since these base lines are used in other places, it is better to compare with them as well.

**Experience Assessment:**

I do not know much about this area.

**Review Assessment: Checking Correctness Of Derivations And Theory:**

I did not assess the derivations or theory.

**Review Assessment: Checking Correctness Of Experiments:**

I assessed the sensibility of the experiments.

**Review Assessment: Thoroughness In Paper Reading:**

I made a quick assessment of this paper.

---

> ### Author Response · Authors · 2019-11-08
> **Reply to Reviewer1**
>
> Dear reviewer, thank you for your time and comments. First, we would like to draw your attention to our  general comment, answering questions about overall baselines for anomaly localization and statistics on the benefits of our method. This comment also explains changes in the last revision of the paper.
>
> We address here each of your concerns:
>
> - « The major concern is how the quality of f_{VAE} is estimated. From the paper it seems f_{VAE} is not updated. Will it be sufficient to rely a fixed f_{VAE} and blindly trust its quality? »
> For a full context, we remind that the VAE is first trained on a dataset comprising only normal data,  to obtain an estimate of the probability distribution of normal data. Since it is a standard VAE training, the quality of the model can be assessed using any of the classical techniques (cross validation, visual inspection, etc). During inference, the underlying VAE model’s weights are indeed frozen and the only optimized parameters are the input image’s pixels, in an adversarial example’s fashion. As you suggest, we could potentially update the underlying model with test data identified by our method as normal, as in a continuous learning setup, but we leave this to future work.
>
> - « Table 1: It is not clear how "the mean improvement rate of 9.52% over all baselines" was calculated. »
> Following your suggestion, we clarified how this metric was calculated, and added a few statistics on the benefits of our method in the last revision. They were computed by aggregating the improvement rate between a baseline and its grad-augmented version $(AUC_{grad} – AUC_{base}) / AUC_{base}$, over all presented baselines and datasets.
>
> - « Figure 3: Will VAE-grad or DASE-grad perform better? Since these base lines are used in other places, it is better to compare with them as well. »
> We augmented figure 3 with the three remaining baselines. They show similar results to the L2AE on these images. Due to the lack of space, we added this comparison in appendix C.
>
> We hope that we have answered your concerns, thank you again for your suggestions.

---

### Official Review · AnonReviewer2 · 2019-10-28
**Official Blind Review #2**

**Rating:** 8

**Review:**

The paper improves anomaly detection by augmenting generative models (VAE, etc) by iteratively projecting the anomalous data onto the learned manifold, using gradient descent of the autoencoder reconstruction term relative to the image input. The work seems related to AnoGAN, only instead of iterating over the latent space, the iteration is over the more expressive input space. The method is intuitive and a good parallel to Adversarial projections is made in the paper. To the best of my knowledge, the idea is novel, although I am not completely sure.
The second idea in the paper is to scale the losses by the reconstruction accuracy, which also is intuitive and shown to significantly speeds up the model convergence.

The experimental results are pretty convincing, showing both quantitatively and qualitatively that the method improves consistently over using the underlying vanilla generative models (AE/DSAE/2 VAEs). One desirable improvement is to get error bounds on the results, those are currently missing. Also, based on the inpainting results in Fig 7, it's not really clear if the method generates better results than Ivanov et al.






**Experience Assessment:**

I have read many papers in this area.

**Review Assessment: Checking Correctness Of Derivations And Theory:**

I carefully checked the derivations and theory.

**Review Assessment: Checking Correctness Of Experiments:**

I assessed the sensibility of the experiments.

**Review Assessment: Thoroughness In Paper Reading:**

I read the paper at least twice and used my best judgement in assessing the paper.

---

> ### Author Response · Authors · 2019-11-08
> **Reply to Reviewer2**
>
> Dear reviewer, thank you for your time and comments. First, we would like to draw your attention to our  general comment, answering questions about overall baselines for anomaly localization and statistics on the benefits of our method. This comment also explains changes in the last revision of the paper.
>
> In particular, in order to better illustrate the variability of the results associated with our method, we added in appendix F a  histogram of the AUC improvement rate on all datasets and architectures reported in table 1.  The median improvement rate over all baselines and datasets is at 4.33%, the 25th percentile at 1.86% and the 75th percentile at 15.86%.
>
> Concerning the inpainting comparision with Ivanov et al., please note that due to the « creative » nature of the inpainting task, a quantitative metric is hard to define. Nevertheless, we added those results as an interesting application of being able to project on a learned manifold, and reproduced the results from Ivanov et al. from their provided model for the sake of a comparison with another VAE-based method. Figure 8 shows that the quality of the reconstructions in both methods is comparable, even though our VAE is trained without any assumptions over the mask's properties. Following your suggestion, we added a comparison sentence in the caption for figure 8.
>
> We hope that we answered your concerns, thank you again for your review.

---

### Author Response · Authors · 2019-11-08
**General comments**

Dear reviewers, thank you all for your time and comments. We have followed your suggestions in our new revision of the paper and we believe it strengthens its content. We note that reviewers were positive about the significance of our work, that « discusses an important problem of solving the visual inspection problem limited supervision », and they note that our general approach is « intuitive », and « [leads] to significantly better results », while our second idea  « significantly speeds up the model convergence ». Nevertheless, several questions are raised on what constitutes the overall baseline for unsupervised anomaly localization, as well as the need for further statistics on the benefits of our method.

- As the authors of Bergmann et al., 2019, we acknowledge the lack of an overall « best » baseline for anomaly localization, but we want to emphasize that our contribution is a method to increase the performance of any autoencoder-based model.

- Thus, to give a better sense of the overall improvements of our method, we computed the histogram of the improvement rate in AUC between a baseline and its grad-augmented counterpart, over all datasets and over all baseline models. We added this histogram and a short analysis in appendix F. We reported additional statistics of this overall improvement distribution in the main text: the median improvement rate over all baselines and datasets is at 4.33%, the 25th percentile at 1.86% and the 75th percentile at 15.86%.

- We clarified the table of results, highlighting the AUC increase or decrease with colors instead of arrows.

- We augmented figure 3 with the three remaining baselines, which shows similar results to the L2AE. Due to the lack of space, we added this comparison in appendix C.

We hope that we answered here most of your concerns. We will also answer each of your comments in detail.

---

### Decision · Program_Chairs · 2019-12-19

**Decision:**

Accept (Poster)

**Comment:**

This paper proposed to use an autoencoder based approach for anomaly localization. The method shows promising on inpainting task compared with traditional auto-encoder.

First two reviewers recommend this paper for acceptance. The last review has some concerns about the experimental design and whether VAE is a suitable baseline. The authors provide reasonable explanation in rebuttal while the reviewer did not give further comments.

Overall, the paper proposes a promising approach for anomaly localization; thus, I recommend it for acceptance.